# Structural and Morphogenetic Characteristics in *Paspalum notatum*: Responses to Nitrogen Fertilization, Season, and Genotype

**DOI:** 10.3390/plants12142633

**Published:** 2023-07-13

**Authors:** Roberto R. Schulz, Alex L. Zilli, Elsa A. Brugnoli, Florencia Marcón, Carlos A. Acuña

**Affiliations:** 1Facultad de Ciencias Agrarias, Universidad Nacional del Nordeste, Corrientes 3400, Argentina; rober.r.schulz@gmail.com (R.R.S.); azilli@agr.unne.edu.ar (A.L.Z.); abrugnoli@agr.unne.edu.ar (E.A.B.); fmarcon91@gmail.com (F.M.); 2Instituto de Botánica del Nordeste, Facultad de Ciencias Agrarias, Universidad Nacional del Nordeste, Consejo Nacional de Investigaciones Científicas y Técnicas, Corrientes 3400, Argentina

**Keywords:** forage ecology, warm-season grasses, N-fertilization

## Abstract

Understanding leaf generation dynamics, their seasonal changes, and their responses to nitrogen fertilization (NF) is key to improving pasture utilization efficiency. The objectives of this research were to determine structural and morphogenetic variables underlying changes in herbage mass on a set of *Paspalum notatum* genotypes. Ten *P. notatum* genotypes were evaluated in experimental plots following a completely randomized block design under a split-plot arrangement for two N-rates during four periods. Increased herbage mass (HM) after N-fertilization was explained by a higher tiller density (TD) (41.8%) and tiller weight (TW) (22.1%). The increment of TW after NF was due to the increase in leaf blade length (LBL) and width (LBW). During the flowering season, NF increases the reproductive tiller density by 262.5%. Seasonal variation in HM was mainly explained by changes in LBL that modified TW. Morphogenetic traits differed between genotypes of different growth habits; therefore, different management practices are suggested. The average increase in leaf elongation rate in response to NF was about 36.7%, generating longer leaves despite reductions in leaf elongation time (LET). The depletion in LBL and consequently in TW and HM during the autumn was attributed to the reduction in LET.

## 1. Introduction

The beef cattle industry is mostly based on the use of natural grasslands and rangelands or planted pastures [1]. In the case of developing countries, naturally occurring grasslands represent an important feed source for cattle [1]. In the last century, lands traditionally used for animal production have been turned into more profitable activities, principally agriculture, and urban areas. This fact forced the cattle industry to become more efficient in the use of land and other resources [2,3,4,5,6]. Adopting more productive forage species and managing practices adapted to the species and productive systems represents a valuable tool for improving animal productivity.

Bahiagrass (*Paspalum notatum* Flüggé) is a perennial, rhizomatous grass species native to America and is the principal component of grasslands in Southern Brazil, Paraguay, Uruguay, and Northeastern Argentina [7]. It is well adapted to heavy grazing, low fertile soils, and poor management. It has been cultivated in several countries around the world, principally in the Southern USA where it represents one of the main forage species for grazing systems [8]. The species has two cytotypes, the apomictic tetraploid, and the sexual diploid.

Plant growth is directly related to light interception and this is to the leaf area index [9]. However, once the canopy exceeds an optimum leaf area index, respiration of the lowest leaves will be greater than photosynthesis, and the growth rate of the whole sward will be lower than at the optimum leaf area index [10]. In a grass forage species, the leaf area index is defined by the combination of three structural variables: tiller density, number of living leaves per tiller, and leaf size [9]. Leaf size and the number of leaves per tiller determine the tiller weight; therefore, herbage mass would be the product of the tiller weight and tiller density of a given sward [11]. In addition, leaf size is easily influenced by air temperature, and hydric and nutritional deficits, whereas the number of living leaves per tiller is mainly defined by the genotype, and scarcely affected by the environment [9].

Morphogenesis is defined as the dynamic generation and expansion of the plant canopy. This variable is determined by the rates of leaf generation, expansion, senescence, and decomposition of different organs of the plant [9]. The structural variables mentioned above, tiller density, leaf size, and the number of leaves per tiller, are affected by leaf appearance rate (or its inverse, the phyllochron), leaf elongation rate, and leaf half-life [9,12]. Leaf appearance rate has an important role in defining tiller density by setting the number of buds, which are potential tillers. The “site filling” is defined as the proportion of tiller generated over the total number of buds. In addition, the interaction of the morphogenetic variables defines leaf size and the number of living leaves per tiller, that then determines the tiller weight [9]. Morphogenetic variables, and therefore, structural variables, are defined by the genotype, but they are also strongly affected by environmental factors such as water content and soil fertility, air temperature, and light quality [9,13,14].

Nitrogen is considered the most limiting nutrient for vegetal growth in the majority of soils around the world [15] and its deficiency is one of the most significant causes of pasture growth reduction in tropical and subtropical regions [16]. In addition, the high response of nitrogen fertilization on forage production by warm-season grasses (C4) in comparison to temperate (C3) ones is due to its greater nitrogen use efficiency [17]. The increase in forage production by nitrogen fertilization has been extensively reported in the last decades [15,18,19,20,21,22,23,24,25]. The response to N-fertilization has been widely studied in *P. notatum*. For instance, Beaty et al. [26] reported an increment in herbage mass from 3.34 to 10.3 Mg ha^−1^ for N-rates of 0 and 270 Kg ha^−1^ respectively. Beaty et al. [27] reported increments of herbage mass for N-rates of 0, 84, and 168 Kg ha^−1^, whereas the higher rates (336 Kg ha^−1^) did not differ from 168 Kg ha^−1^. Allen et al. [28] reported 11.98, 14.34, and 15.66 Mg ha^−1^ for N-rates of 224, 336, and 448 Kg ha^−1^. Burton et al. [29] reported 6.01, 8.24, 11.9, and 15.2 Mg ha^−1^ for N-rates of 56, 112, 224, and 448 Kg ha^−1^, respectively. Therefore, *P. notatum* herbage mass response to N-fertilization is rather linear at moderate N-rates, but less consistent responses are reported for higher rates. However, even though N-fertilization practices have been widely studied in the species, the eco-physiological mechanisms involved in this response remain unclear.

N-fertilization is also related to seed production on the species, and management practices were generated in order to increase seed yield. For instance, Adjei et al. [30] evaluated the effect of burning or mowing at different stages and applying N-rates of 0, 50, and 100 Kg N ha^−1^, reporting that N-fertilization increased seed yield and inflorescence density, but has no impact on seed weight and seed quality. Similarly, Adjei et al. [31] evaluated three managing practices, including 100, 200, and 300 Kg N ha^−1^, reporting a quadratic response of seed yield and inflorescence density to N-rates. Recently, Rios et al. [32] evaluated the timing of defoliation and 0, 60, and 120 Kg N ha^−1^ N-rates in a set of advanced breeding lines, reporting that N-fertilization increased inflorescence density, but did not affect seed quality traits. In addition, the authors found that the optimum N-rate differs according to the growing habits of the genotype, where the forage type (upright genotypes) responded linearly to N-rates, whereas the turf type (prostrated genotypes) responded to 60 Kg N ha^−1^, but a higher rate (120 Kg N ha^−1^) reduced inflorescences density. Determining the effect of N-fertilization over reproductive tiller density in new promising breeding lines is important to determine its impact on seed production in new cultivars.

Previous reports about the effect of nitrogen fertilization on the leaf appearance rate are contradictory. Cruz and Boval [33] reported variations between species, for instance, small changes for this variable were observed in *Setaria anceps* and *Digitaria decumbens*, whereas an increment of 40% was observed for *Dichantium aristatum*. In addition, a significant increase in leaf appearance rate was reported in *Megathyrsus maximus* cv. Mombasa [34], *Urochloa brizantha* cv. Marandú [35], *U. brizantha* cv. Xaraés [36], and in *U. brizantha* and *U. decumbens* [37]. In tall fescue (*Festuca arundinacea*), Gastal et al. [38] attribute the increase in leaf area index after nitrogen fertilization to the increment of the leaf elongation rate. However, taking into account that nitrogen fertilization could increase leaf appearance rate, and that is inversely proportional to leaf elongation time, it is possible that an increment of leaf elongation rate will not result in an increment of leaf size, and therefore, leaf area index [12]. Finally, leaf half-life, a critical variable for grazing management, could be reduced after nitrogen fertilization because of the increment of the leaf appearance rate, accelerating the turnover in order to maintain a constant number of living leaves per tiller.

Warm-season perennial forage species exhibit a reduction in biomass production during the winter, even under conditions of high air temperature and hydric and nutritional availability [26,28,39,40]. Sinclair et al. [41] reported that these species are sensitive to day length; therefore, their growth during the cool season is restricted despite suitable air temperature and hydric and nutritional availability. In *P. notatum*, genetic variation for day-length sensitivity is observed, and the selection of less sensible genotypes allowed the release of cv. UF-Riata, which exhibited a greater extension of the growing season [8]. Low day-length sensitivity is a sought trait in a new cultivar due to allowing a greater grazing period, reducing the cost of animal supplementation. The effects of day length over structural and morphogenetic variables remain scantly known. The understanding of morphogenetic variables involved in the responses of herbage mass and reproductive differentiation of the species to N-fertilization is fundamental to improving forage production and utilization by the formulation of improved management recommendations in addition to the generation of highly productive cultivars in terms of forage and seeds.

The morphogenetic characterization of advanced genetic lines of forage grasses, together with the study of seasonal changes in the morphogenetic and structural variables that determine iPAR (intercepted Photosynthetically Active Radiation), would be key not only in optimizing grazing management but also in the selection process of advanced lines. Zilli et al. [42] selected a group of *P. notatum* apomictic hybrids based on superior agronomic characteristics at individual plant levels and exhibiting different growth habits (relation between height and diameter). More recently, Brugnoli et al. [43] performed an evaluation of seed production and herbage mass at plot level on a set of 13 promising hybrids, reporting differences in terms of seed production; however, no differences in herbage mass were observed between genotypes. An eco-physiological characterization of these superior hybrids, herbage mass seasonal variation, and its response to N-fertilization would be critical for the identification of superior lines for future cultivar releases.

The objectives of this research were: (1) to estimate the difference in herbage mass and its seasonal variation in poor vs. N-enriched environments in a group of apomictic hybrids and cultivars of *P. notatum*; (2) to estimate the structural and morphogenetic variables, its seasonal variation, and its relation with light interception; (3) to determine the structural and morphogenetic variables governing herbage mass differences in *P. notatum* and (4) to characterize reproductive differentiation patterns in response to nitrogen fertilization.

## 2. Results

The effects of N-fertilization, season, and genotype were evaluated over 15 structural and morphogenetic variables listed in Table 1.

### 2.1. Herbage Mass

All main effects were statistically significant, except the genotype. In addition, the two ways interactions between period×nitrogen and period×genotype were significant (Appendix A). Nitrogen increased HM and HAR by 70% on average across genotypes and periods of evaluation (Table 2). The greatest HM was observed for Spr-Sum18-19, and the lowest during the Winter18, for both N-rates. HAR was significantly affected by the period of evaluation, where those periods with higher average temperatures and photoperiod exhibited greater HAR across both nitrogen levels (Table 2). Prostrated genotypes exhibited lower HM and HAR than upright and intermediate ones when comparisons were performed between genotypes grouped by their growth habits (Table 3).

### 2.2. Tiller Density and Weight

Tiller density and tiller weight were significantly affected by all main effects in addition to the two-way interaction period×nitrogen and period×genotype (Appendix A). Nitrogen fertilization increased TD by 41.8%, from 972 to 1378 tiller m^−2^, and TW by 22.1%, from 221.3 to 270.3 mg tiller^−1^. A significant increment in TD was observed throughout the periods, meanwhile, the greatest TW was observed in those periods with higher temperatures and photoperiod (i.e., Spr-Sum18-19). Prostrated genotypes exhibited greater TD and lower TW in comparison to upright and intermediate genotypes. Lines I7 and C11 showed the lowest TD, and line F44 the lowest TW throughout the periods. On the other side, the greatest TD was exhibited by lines I21, F44, B7, B14 and cv. Argentine and Boyero, and the greatest TW by C14, I7, G37, B14, B7, C11 and cv Boyero (Table 3).

### 2.3. Leaf Blade Size

The period of evaluation, nitrogen fertilization, and genotype had a significant effect on LBL and LBW, in addition to the two-way interaction period×nitrogen and period×nitrogen (Appendix A). Nitrogen significantly increased LBL and LBW by 22% and 7% respectively, across periods and genotypes (Table 2). Regarding the period of evaluation, shorter and wider leaves were observed for Winter18, whereas longer and thinner leaves were observed during Spr-Sum18-19 (Table 2). Prostrated genotypes exhibited shorter and wider leaves in comparison to upright and intermediate ones. Cultivar Boyero and lines C14, I7, B7, and B14 exhibited longer leaves than cultivar Argentine and line F44. The greatest LBW was observed for line C11 and the lowest for I21 and cv. Boyero (Table 3).

### 2.4. Number of Leaves per Tiller

Period of evaluation and genotype had significant effects over NTL, NEL, and NUL, in addition to the two-way interaction nitrogen×period for NUL (Appendix A). The lowest NTL per tiller was observed during Spr-Summ18-19, which is the period with the highest temperatures. Concurrently, NEL and NUL were the lowest in Spr-Sum18-19 and the greatest in those periods with lower temperatures, Winter18 and Sum-Aut19 (Table 2). The greatest NTL and NEL were observed by line F44, whereas the lowest was by I7. NUL exhibited a low variation between lines, where lines G37, I21, F44, B14, and cv. Argentine showed the greatest values, meanwhile, lines B7, I7 and C14 exhibited the lowest. Prostrated genotypes exhibited greater NTL, NEL, and NUL than upright genotypes (Table 3).

### 2.5. Phyllochron

Phyllochron was significantly affected by the period of evaluation, nitrogen, genotype, and the two-way interaction genotype×period (Appendix A). The greatest values were observed for Spr-Sum18-19 (200.6 °Cd), followed by Winter18 (175.6 °Cd) and finally Sum-Aut19 (144.7 °Cd), across nitrogen levels (Table 2). N-fertilization reduced the Phyll. by 10.2% (from 180 to 165.2; Table 2). Regarding genotypes, lines G37 and C14 showed the greatest values, whereas line F44 and cv. Argentine were the lowest. On average, a lower amount of thermal sum between leaves was required for prostrated lines (147.8 °Cd) than upright and intermediate genotypes (187.8 °Cd and 181.8 °Cd, respectively) (Table 3).

### 2.6. Leaf Blade Dynamics

Period of evaluation and genotype significantly affected LER, LHL, and LET, whereas nitrogen affected LER and LET, but not LHL. Two-way interaction period×nitrogen was significant for LHL (Appendix A). Nitrogen fertilization increased LER by 36.7% (from 0.6 to 0.82 mm °Cd^−1^), whereas reduced LET by 10.1% (from 370.1 to 332.5 °Cd). Regarding the period of evaluation, LET and LHL were significantly greater for Spr-Sum18-19, whereas a greater LER was observed for Sum-Aut19 at 0 N-level in comparison to Spr-Sum18-19 (Table 2). The greatest LER was observed for lines C14, I7, B7, and cv Boyero. Those lines with upright growth habits exhibited greater LER than intermediate and prostrate ones. Comparably, the greatest LHL was observed for lines C14, I7, B7, G37, and cv Boyero. Upright and intermediate growth habit lines exhibited greater LHL than prostrated ones. Concerning LET, lines G37 and B14 exhibited the greatest values, whereas lines F44, I21, and cv Argentine were the lowest. The greater LHL was observed for upright and intermediate lines in comparison to prostrated ones (Table 3).

### 2.7. Correlation among Variables

Pearson’s correlation analyses revealed that HM and HAR were strongly correlated with TW (0.91), LBL (0.76), LHL (0.58), and TD (0.53). Likewise, TW was strongly correlated with LBL (0.7) and LHL (0.68), whereas LBL was correlated with LER (0.64) and LHL (0.56). In addition, LHL was correlated with Phyll (0.57). Phyllocron were positively correlated with LET (0.8) and LHL (0.57), and negatively with NEL (−0.6) and NTL (−0.58). The LET was positively correlated with Phyll (0.8) and LHL (0.58), and negatively with NTL (−0.55) and NEL (−0.53).

Prostrated genotypes grouped on the negative side of PC1 in the principal component analysis were associated with a greater number of leaves (NTL, NEL, and NUL) and tiller density. Upright genotypes grouped at the positive side of PC1 and the positive side of PC2 were characterized by a greater HAR, LBL, LER, LHL, and TW. Genotypes of intermediate growth habit grouped mainly at the negative side of PC2 were characterized by a greater LBW, Phyll, and NEL (Figure 1).

### 2.8. Light Interception

During the period Winter18, the average light interception was significantly increased by N-fertilization at days 27, 55, 67, 77, 116, and 127 after harvest. However, iPAR remained at a level below 60% during the entire period for both, fertilized and non-fertilized plots (Figure 2a). For Spr-Sum18-19 a greater iPAR was observed for fertilized plots during the complete period, except for the first measurement performed 7 days after harvest. The favorable temperatures and precipitation registered during the period (Figure 3) allowed 91% of iPAR for fertilized plots between 46 and 52 days after harvest, and 74% for non-fertilized plots at day 98 after harvest (Figure 2b). It is remarkable that precipitation registered during Spr-Sum18-19 and Sum.Aut19 was superior to the normal for the regions; however, because of the well-drained soil, non-water related stresses (drought or waterlogging) were observed and conditions were adequate for plant growth during the full length of the experiment. Similarly, during the period Sum-Aut19 fertilized plots exhibited a greater iPAR from 28 days after harvest and reached 90% between 39 and 55 days after harvest, whereas non-fertilized plots showed 74% at the end of the period (Figure 2c). Finally, the evolution of the iPAR during the period Win-Spr19 was slow for the coolest part of the period averaging 35.6% and 19.8% for fertilized and non-fertilized plots, respectively, at 102 days after harvest; whereas iPAR rapidly increased as temperatures increased, reaching 58.44% and 50.2% for fertilized and non-fertilized plots. Significant differences were observed for the measurement performed 40, 46, 88, 95, 102, 131, and 151 days after harvest (Figure 2d).

### 2.9. Tiller Differentiation

During the reproductive period of the species, Spr-Sum18-19, nitrogen fertilization significantly increased total tiller density by 50% (from 979.5 to 1469.9 tiller m^−2^). In addition, reproductive tiller density (RTD) was increased by 257.7% (from 143.2 to 512.3 tiller m^−2^), whereas vegetative tiller density (VTD) was not significantly affected by nitrogen fertilization (957.6 and 836.3 tiller m^−2^ for fertilized and non-fertilized, respectively). Nitrogen fertilization increased the reproductive differentiation (proportion of reproductive tillers out of the total tillers) by 137.5% (from 14.4% to 34.2%). A great variation was observed for RTD between genotypes, in addition to a significant effect of the two-way interaction genotype×nitrogen (*p* = 0.012). The greater RTD for fertilized plots was observed for lines I7, I21, and B14, in addition to C14 for non-fertilized plots. On the other side, lines G37 and B7 exhibited the lowest values for both fertilization treatments (Figure 4).

## 3. Discussion

The knowledge of leaf area generation in grasslands or pastures is a key tool to optimize the use and management of forage resources, aiming to increase production and utilization efficiency without affecting sward persistence. Therefore, the characterization of structural and morphogenetic variables of a pasture, together with its response to seasonal variations and practice management such as nitrogen fertilization, are key for improving grazing practices, aiming to maximize iPAR, and forage production and quality, maintaining an adequate balance between growth, senescence, and herbage consumed by animals [9]. In this research, we evaluated the herbage mass and distribution throughout the seasons and characterized a group of new apomictic lines and cultivars for their structural and morphogenetic traits.

The herbage mass of warm-season species is concentrated in the warmest months. In this study, we observed that 64.7% of the yearly herbage production was generated during spring and early summer, 23.7% during late summer and fall, and only 11.6% during the winter. This pattern is similar to that reported by Beaty et al. [39] for diploid cultivars of the species, where 63.7% of the total herbage mass was produced during the three warmest months of the year. In addition, Hirata et al. [44] reported similar results in a set of cultivars of *P. notatum*. Sinclar et al. [41] evaluated the forage growth of warm-season forage species during the cool season, reporting that the reduction in forage production is attributable to short-day length. Variation for day-length sensitiveness between genotypes was observed in this study. For instance, cv Argentine produced 72% of the year-round herbage mass during late spring and early summer, whereas lines I7 and F44 exhibited 60.9% and 60.4%, respectively, for the same period. HM during Winter18 was significantly different for the lines mentioned above in comparison with the cv Argentine, whereas both lines did not differ from cv Argentine for Spr-Sum18-19 and Sum-Aut19. Extending forage production throughout the year is a sought trait for grazing-based production systems. Hence, the apomictic breeding lines used in this research were previously selected based on a set of agronomic variables including cold tolerance and superior cool-season growth [42,43]. This result indicates that the breeding succeeded in increasing forage biomass production during the cold season without decreasing the warm season production. Due to the relevance of HM distribution throughout the year by warm-season species, HM produced during the cool season should be included as a selection variable in breeding programs.

Regarding N-fertilization, a significant increment of 78.6% in herbage mass was observed; however, seasonal distribution was not affected. Allen [28] observed a significant increase in herbage production after N-fertilization, but its seasonal distribution remained almost unchanged. On the other hand, the impact of N-fertilization was variable according to the season. For instance, N-fertilized plots showed 100% higher HM than non-fertilized during Sum-Aut19 and Win-Spr19, whereas during Spr-Sum18-19 and Winter18 HM was 72% and 45% greater for N-fertilized plots, respectively. Therefore, N-fertilization would be of great impact on increasing forage production for grassland farming, although other practices, such as the confection of forage reserves based on forage produced during the warm season would be necessary to overcome the winter forage gap in the subtropics. Non-significant differences in herbage mass were observed between genotypes; however, it is remarkable that all of them produced as much herbage mass as the most popular tetraploid cultivar of the species.

The forage production components on a sward are the tiller density and the tiller weight, and both variables are usually negatively correlated [9]. Moreover, the number of living leaves per tiller, and the size of these leaves define tiller weight. Therefore, an increment in biomass production in response to N-fertilization could be due to an increment in tiller density and/or tiller weight. In our research, we observed a non-significant correlation between TD and TW, but these variables were highly correlated with HM by 0.53 and 0.91, respectively. In addition, tiller density was significantly increased by N-fertilization by 41.8%. Graminho et al. [45] reported similar results for diploid and tetraploid genotypes of *P. notatum*, reporting increments from 30% to 87% in tiller density using an N-rate of 480 Kg N ha^−1^. Pakiding and Hirata [46] reported a significant increment of tiller density after N-fertilization only using a stubble height below 12 cm. In addition, tiller density increased throughout the periods of evaluation. Growth habits had a significant effect on TD, where greater values were observed for prostrated genotypes than upright and intermediate. Interrante et al. [47] reported greater TD on cv Argentine, a prostrated genotype, than diploid cultivars, characterized by upright habits. In addition, the authors reported an increment of TD under high defoliation pressure on upright genotypes. The higher TD in prostrated genotypes and genotypes under intensive defoliation management could be related to a deeper light canopy penetration, reaching growth points, and promoting site filling [9]. This statement is supported by the fact that prostrated genotypes exhibited lower iPAR than the upright ones.

In this study, N-fertilization increased tiller weight by 24%, across genotypes and periods. Regarding NTL, a great variation between genotypes was observed, from 5.6 to 8.5, with lower values for upright genotypes and higher for prostrated ones. However, no effect of N-fertilization was observed. Therefore, the increase in tiller weight could be a consequence of greater leaf size since a significant increment in LBL was observed after N-fertilization in all evaluated periods, except Winter18, averaging 22.2% across periods, and reaching 54% during Spr-Sum18-19. This is supported by the high correlation found between tiller weight and LBL (0.7). This behavior was not as high for LBW, where only a 7.1% increment was observed. These results are in agreement with those reported by Pakidin and Hirata [46] for *P. notatum* cv Pensacola, and Cruz and Boval [33] for *Dichanthium aristatum* and *Digitaria eriantha*.

The period of evaluation played an important role in the number of leaves per tiller. The lowest number of leaves was observed for the warmest period (Spr-Sum18-19), whereas Winter18 and late summer and autumn (Summ-Aut19) exhibited a greater number of leaves. This pattern is contrary to the one observed for tiller weight, hence, despite the lesser number of leaves per tiller for Spr-Sum18-19 these leaves over-compensate with a greater size. Moreover, LBW during the warmest period was 31% lower in comparison to Winter18. Therefore, the greater LBL observed for the warmest period could explain the greater leaf size and TW. For instance, LBL in Spr-Sum18-19 was 131% higher than in Winter18. Hirata and Pakiding [11] and Pakiding and Hirata [48] previously reported a greater leaf size during the spring.

The phyllochron and the leaf half-life define the number of leaves per tiller. Chapman and Lemaire [9] stated that these variables are not affected by nitrogen availability; however, there is contradictory research on this topic. For instance, a significant reduction in phyllochron after N-fertilization was reported in wheat (*Triticum aestivum*) [49], in *Dactylis glomerata* [50,51], in *Megathyrsus maximus* cv. Mombasa [34], and in *U. brizantha* cv. Marandú [35]. Pakiding and Hirata [52], and Islam and Pakiding [53] reported a thin reduction of the phyllochron in *P. notatum* var Pensacola. In agreement with Chapman and Lemaire [9], in this study small or non-significant differences were observed in Phyll and LHL after N-fertilization; however, a shorter Phyll was observed for Sum-Aut19 for both N-rates. Apparently, the effect of N-fertilization is species-dependent, and the magnitude is affected by the season of evaluation, suggesting that other conditions, such as temperature and photoperiod, could be involved in this response.

Regarding the period of evaluation, greater phillochron was observed for those leaves generated during the warmest period, Spr-Sum18-19 in comparison to Sum-Aut19 and Winter18, whereas the inverse behavior was observed for LHL. In contrast, Egger et al. [54] reported an average phyllochron of 157, 165.5, and 345 °Cd^−1^ during the spring, summer, and autumn, respectively, for *P. notatum* F_1_ hybrids. However, these discrepancies could be related to the fact that in this study, the criteria for defining periods were not seasons but leaf senescence; hence, a clear comparison with the previous work is not possible. The lower NTL observed for Spr-Sum18-19 indicates that the longer LHL for this period was not enough to maintain a greater number of leaves per tiller due to a greater phyllochron, i.e., a lower leaf appearance rate. In addition, great variation was observed for these traits between genotypes. It is remarkable that prostrated genotypes exhibited greater NTL and shorter phyllochron, whereas upright genotypes showed contrary behavior.

The leaf half-life differs between species and genotypes and its determination is key for appropriate forage sward utilization, because it represents the period between a harvest and the point of maximum foliar biomass accumulation without leaf senescence, allowing for a suitable reserve level in the plant [12]. Lemaire and Agnusdei [12] reported an LHL of 570, 377, and 745 °Cd in C_3_ grasses for *Festuca arundinacea*, *Lolium multiflorum*, and *Stipa neesiana*, respectively. In this study, we observed an average LHL of 1126.4, varying between 1188 and 1052.6 °Cd for cvs Boyero UNNE and Argentine, respectively. In general, a shorter LHL was observed for prostrated genotypes indicating a faster leaf turnover, whereas intermediate and upright genotypes exhibited the longest. Therefore, prostrated genotypes should be managed with higher harvest frequencies than upright and intermediate ones in order to obtain greater harvest efficiency and animal performance, whereas the upright and intermediate ones would need a longer harvest gap to foment a greater sward persistence. Moreover, LHL was greatest during the warmest period of evaluation; however, in terms of chronological time (d), leaf senescence would start with a lower number of days during the warm season in comparison with the cold season and consequently, harvest frequencies should be adjusted. For instance, in terms of chronological time, Pakiding and Hirata [52] and Pakiding and Hirata [55] reported an LHL of 108 to 203 d during the autumn and 39 to 69 d during the spring in *P. notatum* cv Pensacola.

Reports about the effect of N-fertilization on LHL are controversial. A significant reduction in LHL after N-fertilization was reported in *Urochloa. brizantha* cv. Xaraés [36], in *U. brizantha* and *U. decumbens* [37] and in *Axonopus aureus* [56], whereas the contrary effect was reported in *M. maximus* cv. Mombasa [34]. Moreover, no effect was observed in *M. maximus* cv. Tanzania [57], and in *P. notatum* cv. Pensacola [52]. In our research, we observed no effect of N-fertilization on LHL.

As was previously mentioned, tiller weight was significantly increased by N-fertilization, and this response was mainly caused by an increment of LBL. The greatest LBL observed in fertilized plots could be explained by an increment of the LER and/or LET. Our research indicates that N-fertilization increased LER by 36.7%, whereas LET was reduced by 10.1%. Therefore, the increment in LER overcomes the reduction in the time gap where a leaf is elongating, arising longer leaves on fertilized plants. The increment of the LER after N-fertilization has been previously reported in *Lolium tumulentum* [58], *Festuca arundinacea* [38,59], *Dactylis glomerata* [51], and *Megathyrsus maximus* cv. Mombasa [34]. In addition, Hirata [60], and Pakiding and Hirata [48] reported a significant increment of LER in *P. notatum* cv. Pensacola. Regarding genotypes, a greater LER and LET were observed for upright genotypes than intermediate and prostrated ones, consequently exhibiting a greater LBL.

Pearson’s correlation analysis revealed that the greater HM observed for N-fertilized plots is explained by a greater TW, and to a lesser extent to TD. Moreover, the greater LBL was the main responsible for the increment in TW. The greater LBL in fertilized plots accounted for a greater LER, which is characterized as a sensible variable to environmental changes [9]. In the PCA it is possible to observe that the most productive genotypes, upright genotypes in addition to an intermediate one (C14), grouped at the positive sector of PC1 to which most of these variables were associated. On the other hand, the less productive genotypes, prostrated ones, were grouped in the negative sector of PC1.

During the regrowth of a sward, the leaf area index increases allowing a greater iPAR and growth rate, until the leaf area index achieves an optimum iPAR (95%) where the growth rate reaches a plateau and begins to decrease [61]. Moreover, this optimum iPAR is frequently used as a criterion for determining grazing moment. The greater TD and leaf size of N-fertilized plots showed a higher iPAR than the observed for non-fertilized ones. N-fertilization increased the leaf area index allowing for an earlier light interception fraction than non-fertilized plots and leading to a greater herbage mass accumulation. The greatest iPAR was observed during the warmer evaluated periods, where the plants exhibited the greatest LBL and HM. During the cooler periods, iPAR remained below 60% and 40% for fertilized and non-fertilized plots, respectively, which is expected due to the low growth exhibited by the species during the cool season. However, the lower competitiveness of the species during this period is key for the success of over-seeding cool-season species, such as annual ryegrass, small grains, and clovers, in order to increase the productivity of the pasture during a broader period and obtaining other ecosystem services [62].

Upright genotypes exhibited lower tiller density and number of leaves per tiller; higher HM, Phyll., LHL, LER, and LET, leading to heavier tillers formed by a lower number of longer leaves with lower turnover; these characteristics would be desirable for not limiting environments (adequate soil fertility, water content, temperatures, and grazing management). However, under limited conditions, such as intense grazing, hard winter and prolonged drought periods, prostrated genotypes would probably show better tolerance to this kind of stress given that they exhibit 10% more TD, 15% more number of leaves per tiller with faster turnover, and therefore, exposed to intense defoliation, a faster recovery of the leaf area index can be expected. Due to the growth habit, this leaf area will be located closer to the soil level, avoiding herbivory. In addition, prostrated genotypes exhibit a higher accumulation of reserves (root + rhizomes) [63], offering an additional advantage on leaf area index recovery.

Finally, N-fertilization increased reproductive differentiation in all the evaluated genotypes. The effect of N-fertilization over the increment of reproductive tiller density was previously reported for *Panicum coloratum* var. makarikariense [64], *Setaria sphacelata* cv. Nandi [65], *Dichanthium annulatum* [66], and *P. notatum* cvs. Pensacola and Argentine [30,31]. In addition, Bertoncelli et al. [67] reported that N-fertilization significantly increased reproductive tiller density and seed yield; however, the proportion of reproductive tiller over total tillers remained unchanged. In our study, we observed that the relation between the reproductive tillers and total tillers increased from 14.6% to 35% after N-fertilization; therefore, nitrogen plays an important role in the reproductive differentiation of the species. Nevertheless, managing practices should be adjusted according to the objective of the pasture because a higher inflorescence density resulting from N-fertilization will affect the leaf/stem ratio, affecting the forage nutritive value. In addition, the fact that several lines showed a greater RTD than the cultivars used as checks and the great variation observed for this trait between genotypes will allow for the selection of those lines with higher reproductive tiller density for future cultivar releases. Due to the relevance of seed production for forage cultivars adoption [68], further research is needed in order to determine the main components underlying seed yield changes in *P. notatum* and the role of N-fertilization over these components.

In conclusion, N-fertilization significantly increased HM and HAR in all the evaluated periods and genotypes in response to a higher iPAR. The greater iPAR observed on N-fertilized plots was the result of a higher tiller density (constant between periods) and tiller size (variable between periods). N-fertilization did not affect the number of leaves per tiller nor the LHL. A reduction in LET was observed in N-fertilized plots, but the greater increase in LER over-compensates the lower LET allowing a greater LBL (leaf size), and therefore, a greater tiller weight. The seasonal changes observed for the HM and iPAR were related to the lower tiller weight observed due to shorter leaves and not because of changes in the number of leaves per tiller. Grazing management would differ according to growing habits, on one side, prostrated genotypes exhibited higher tiller density, and number of leaves per tiller; lower HM, Phyll., LHL, LER, and LET, leading to lighter tillers formed by a greater number of smaller leaves of faster turnover, whereas upright genotypes exhibited a contrary behavior. Therefore, prostrated genotypes should be grazed using a higher frequency with lower stubble height, in contrast to upright genotypes that will need a longer interval between defoliations and taller stubble height. In addition, an N-enriched environment promotes the reproductive differentiation of tillers, whereas, under a pour environment, a greater proportion of tillers will remain vegetative, probably as a strategy to save resources and boost tiller survival. This study reveals the eco-physiological mechanisms involved in the herbage mass changes of the species which could be extrapolated to the warm-season perennial species. Further research is needed to elucidate if these mechanisms are common under other contrasting environments, such as drought vs. irrigated conditions. Finally, this information could assist the selection process of these promising lines for future cultivar releases.

## 4. Materials and Methods

### 4.1. Experimental Design and Plant Material

The experiment was carried out at the experimental field of the Universidad Nacional del Nordeste, located near the city of Corrientes, Argentina (27°28′ S, 58°47′ W). The soil type was classified as Alfic Udipsamment characterized as sandy soil (93.2% sand, 4.2% slit, and 2.6% clay), well-drained, with a 0.7% land slope, and 130 cm of effective rooting zone. At the beginning of the study, mean soil pH was 6.79, 1.08% of soil organic matter, Bray-I extractable P, and total nitrogen was 10.4 and 500 mg Kg^−1^, respectively. The field was under a perennial grass pasture for four years prior to the beginning of this study, without the addition of fertilizers.

Plant material consisted of eight experimental lines (named F44, I21, I7, B7, C11, C14, G37, and B14) of tetraploid *Paspalum notatum* obtained by Zilli et al. [42], and cultivars Argentine (USDA PI 148996) and Boyero UNNE [69] as checks. Lines and cultivars were previously classified as apomictic [42,69,70], which ensures genetic uniformity among experimental units. Seeds of the ten genotypes were scarified using 98% sulfuric acid for 10 min, washed, and germinated in potting mix. Seedlings were transplanted to 150-mL cell seedling flats. After 30 d plugs were planted to the field on 2 × 2 m plots on 30 November 2017. Plugs were allowed to grow until full plot cover before the beginning of the experiment. The experimental design was a randomized complete block design in a split-plot arrangement with three replications. Main plots (2 × 20 m) consisted of two N levels (0 and 367 Kg N ha^−1^) and sub-plots (2 × 2 m) were the different genotypes. The nitrogen source utilized was urea (46-0-0) and N-fertilization was split into three applications as follows: 67 Kg N ha^−1^ on 03/13/2018 and 150 Kg N ha^−1^ on 11/03/2018 and 02/18/2019. The experiment was conducted under rainfed conditions due to precipitation registered was sufficient for adequate plant growth (Figure 3).

Genotypes were grouped according to their growth habits as prostrated (lines F44 and I21, and cv Argentine), upright (lines I7 and B7, and cv Boyero UNNE), and intermediate (lines C11, C14, G37 and B14).

Mean rainfall, max (TM), and min (Tm) air temperature were registered using an automatic weather station located in the experimental field of the Universidad Nacional del Nordeste (Figure 4). Thermal time (TT) was calculated using the following formula:TT (°Cd) = (TM + Tm)/2 − Tb
where Tb is the base temperature, and represents the temperature at which the vegetative growth stops. Pakiding and Hirata [55] reported that *P. notatum* var. Pensacola interrupts its growth at 7.6 °C; therefore, this value was used as the base temperature for this study.

### 4.2. Evaluated Variables and Data Collection

Response variables were evaluated within plots during four periods. After the full cover, plots were mowed at 5 cm stubble height on 04/26/2018. The evaluated periods were: (a) Winter18 (from 04/26/2018 to 11/01/2018); (b) Spr-Sum18-19 (from 11/01/2018 to 02/14/2019); (c) Summ-Aut19 (from 02/14/2019 to 06/20/2019); (d) Win-Spr19 (from 06/20/2019 to 11/28/2019). Five tillers per plot were tagged a day after harvest to evaluate leaf traits and their dynamics, and the criteria adopted to determine the harvest date was the occurrence of at least one senescent leaf on the tagged tillers.

The herbage mass (HM g DM m^−2^) was measured by harvesting 1 m^−2^ on the center of each plot at 5 cm stubble height at the end of each period. A sub-sample of 300 g of fresh biomass was dried at 60 °C for 72 h until constant weight to determine dry matter concentration. The herbage accumulation rate (HAR mg DM °Cd^−1^) was estimated by dividing HM by the sum of thermal time during the evaluated period.

A day before herbage harvest, tiller density (TD) was determined as the total number of tillers inside of a 0.25 × 0.25 m frame and extrapolated to 1 m^−2^. In addition, during the period Spr-Sum18-19, total tiller density was discriminated between reproductive tiller density (RTD) as the total number of reproductive tillers (tillers with inflorescence), and vegetative tiller density (VTD). Tiller weight (TW) was estimated by dividing HM by TD.

Five tillers per plot were tagged a day after harvest to evaluate leaf traits and their dynamics. The evaluated variables were the leaf blade length (LBL), the leaf blade width (LBW), the total number of leaves (NTL), the number of elongating leaves (NEL), and the number of unfolded leaves (NUL). The LBL was determined as the average length of the last unfolded leaf of tagged tillers, the LBW as the average width of the last unfolded leaf of tagged tillers, the NTL as the average number of leaves of tagged tillers, the NEL as the average number of elongating leaves of tagged tillers and the NUL as the average number of unfolded leaves of tagged tillers.

In addition, morphogenetic variables were determined in each tagged tiller three times per week. Phyllochron (Phyll) was calculated as the accumulated thermal time between the appearances of consecutive leaves (TT from visible leaf tip of consecutive leaves in the same tiller), the leaf elongation rate (LER) as leaf elongation per unit of thermal time (LBL divided by the LET), the leaf half-life (LHL) as the accumulated thermal time from the appearance of a leaf to half-leaf senescence of the same leaf, and the leaf elongation time (LET) as the accumulated thermal time from leaf appearance to fully unfolded (TT from visible leaf tip to visible ligule).

Finally, interception of Photosynthetically Active Radiation (iPAR) was measured weekly using a ceptometer Cavadevices^®^ Bar Rad-100 at noon, avoiding cloudy days.

### 4.3. Statistical Analysis

The statistical analysis was performed using the software Info-Stat [71] linked to the R Studio platform and fitting mixed linear models where genotype, nitrogen, period of evaluation, and their interactions were considered as fixed effects, whereas blocks were considered as random effects. For reproductive tiller density analysis, genotype, nitrogen, and its interaction were considered fixed effects, whereas blocks were considered random effects. Means comparisons were performed using the Fisher’s protected LSD test [72] at *p* = 0.05, performing a correction of error type I [73]. A Principal Component Analysis (PCA) was calculated using mean values of each genotype across four periods and two N-rates. The Pearson correlation coefficient was also calculated.

## Figures and Tables

**Figure 1 plants-12-02633-f001:**
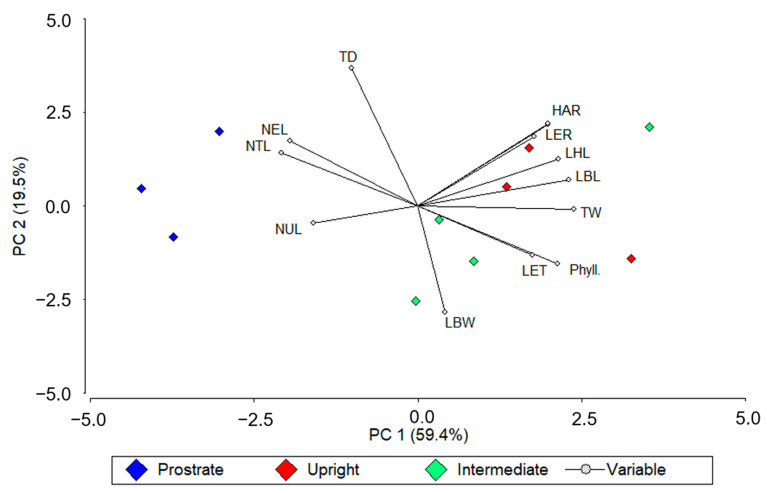
Principal component analysis (PCA) of 12 morphogenetic traits measured on ten genotypes across two N-fertilization rates (0 and 367 Kg N ha**^−^**^1^), and four periods of evaluation. Different colors represent growing habits. Three *Paspalum notatum* genotypes were pooled as “prostrated” (blue symbols), three as “upright” (red symbols), and four as “intermediate” (green symbols). Measured traits were herbage accumulation rate (HAR), tiller density (TD), tiller weight (TW), leaf blade length (LBL), leaf blade width (LBW), number of total leaves (NTL), number of elongating leaves (NEL), number of unfolded leaves (NUL), leaf elongation rate (LER), phyllochron (Phyll.), leaf elongation time (LET), and leaf half-life (LHL).

**Figure 2 plants-12-02633-f002:**
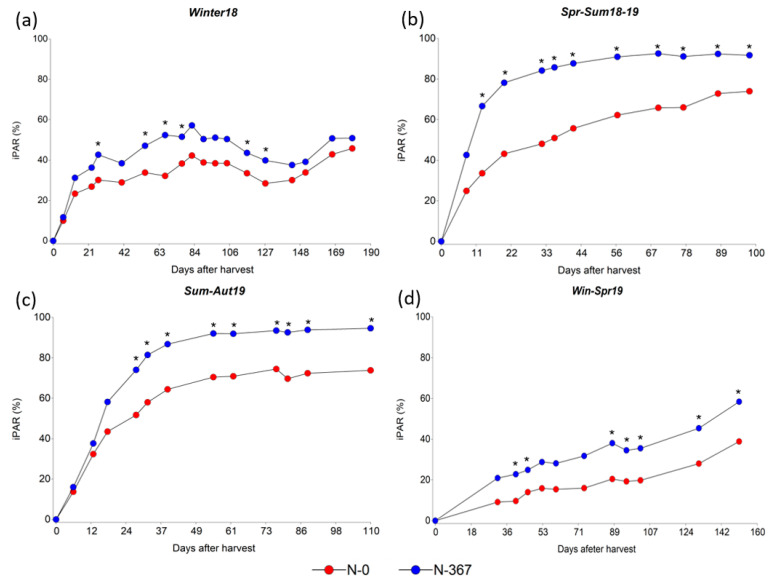
Average iPAR (%) for two N-fertilizations rates, 0 (red symbols) and 367 (blue symbols) kg N ha**^−^**^1^ across ten *Paspalum notatum* genotypes. (**a**) Period Winter18 from 04/26/2018 to 11/01/2018; (**b**) Period Spr-Sum18-19 from 11/01/2018 to 02/14/2019; (**c**) Period Sum-Aut19 from 02/14/2019 to 06/20/2019; (**d**) Period Win-Spr19 from 06/20/2019 to 11/28/2019. An * indicates statistical differences by LSD’s test (*p* < 0.05) among N rates within each period of evaluation.

**Figure 3 plants-12-02633-f003:**
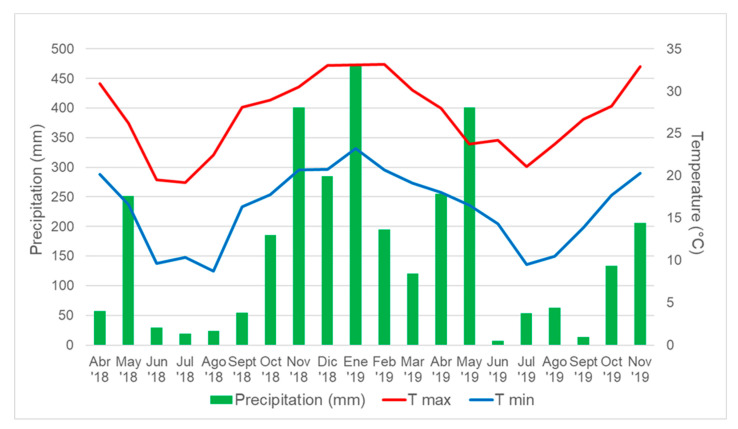
Monthly precipitation (mm) and maximum (T max) and minimum (T min) air temperature (°C) at 1.5 m for the entire period of evaluation at Corrientes City (27°28′ S, 58°47′ W).

**Figure 4 plants-12-02633-f004:**
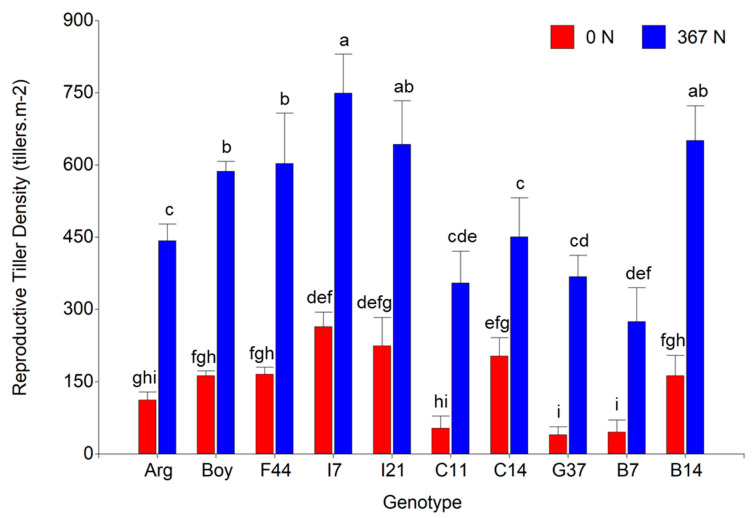
Reproductive tiller density (tiller m**^−^**^2^) for ten *Paspalum notatum* genotypes evaluated for two N-fertilization rates, 0 (red columns) and 367 (blue columns) kg N ha**^−^**^1^. Different letters indicate significant differences for the Fisher’s protected LSD test at *p* = 0.05. Bars represent the standard error.

**Table 1 plants-12-02633-t001:** List of analyzed variables, acronyms used, and units.

Variable	Acronym	Units
Herbage mass	HM	gDM m^−2^
Herbage accumulation rate	HAR	mgDM °Cd^−1^
Tiller density	TD	Nº m^−2^
Tiller weight	TW	mgDM tiller^−1^
Leaf blade length	LBL	cm
Leaf blade width	LBW	cm
Number of total leaves	NTL	Nº
Number of elongating leaves	NEL	Nº
Number of unfolded leaves	NUL	Nº
Phyllochron	Phyll	°C leaf^−1^
Leaf elongation rate	LER	mm °Cd^−1^
Leaf half-life	LHL	°Cd
Leaf elongation time	LET	°Cd
Vegetative tiller density	VTD	Nº m^−2^
Reproductive tiller density	RTD	Nº m^−2^

**Table 2 plants-12-02633-t002:** Means of herbage mass (HM), herbage accumulation rate (HAR), tiller density (TD), tiller weight (TW), leaf blade length (LBL), leaf blade width (LBW), number of total leaves (NTL), number of elongating leaves (NEL), number of unfolded leaves (NUL), phyllochron (Phyll.), leaf elongation rate (LER), leaf elongation time (LET), and leaf half-life (LHL), evaluated over 10 *Paspalum notatum* genotypes during four periods (Winter18 [04/26/2018 to 11/01/2018], Spr-Sum18-19 [11/01/2018 to 02/14/2019], Sum-Aut19 [02/14/2019 to 06/20/2019], Win-Spr19 [06/20/2019 to 11/28/2019]) under two nitrogen fertilization rates (0 and 367 kg·N·ha^−1^).

N Rate	Period	HM	HAR	TD	TW	LBL	LBW	NTL	NEL	NUL	LER	LET	Phyll.	LHL
		g m^−2^	mg m^−2^ °Cd^−1^	Tiller m^−2^	mg tiller^−1^	cm		Leaves tiller^−1^			mm °Cd^−1^	°Cd^−1^		
367	Winter18	122.8 ^e†^	58.0 ^f^	1094 ^c^	114.7 ^f^	13.84 ^d^	0.66 ^a^	7.70 ^a^	4.82 ^b^	2.88 ^a^			172.3 ^b^	1073 ^b^
	Spr-Sum18-19	741.8 ^a^	365.9 ^a^	1470 ^ab^	509.3 ^a^	38.52 ^a^	0.51 ^c^	6.41 ^b^	4.35 ^c^	2.06 ^bc^	0.80 ^a^	368.0 ^b^	187.5 ^b^	1286 ^a^
	Sum-Aut19	298.4 ^c^	158.1 ^c^	1424 ^b^	211.1 ^d^	24.14 ^b^	0.6 ^e^	7.60 ^a^	5.67 ^a^	2.13 ^b^	0.83 ^a^	297.1 ^d^	135.8 ^d^	966 ^c^
	Win-Spr19	371.7 ^b^	182.3 ^b^	1521 ^a^	246.2 ^c^	25.88 ^b^	0.62 ^bc^							
Average 367		383.7 ^A‡^	191.1 ^A^	1378 ^A^	270.3 ^A^	25.6 ^A^	0.6 ^A^	7.24 ^A^	4.89 ^A^	2.36 ^A^	0.82 ^A^	332.5 ^B^	165.2 ^A^	1108 ^A^
0	Winter18	85.2 ^f^	40.2 ^g^	926 ^e^	94.65 ^g^	13.75 ^d^	0.61 ^bc^	7.35 ^a^	4.56 ^bc^	2.79 ^a^			178.8 ^b^	1047 ^b^
	Spr-Sum18-19	431.6 ^b^	212.9 ^b^	979 ^de^	442.7 ^b^	25.15 ^b^	0.45 ^f^	6.64 ^b^	4.50 ^bc^	2.14 ^b^	0.56 ^c^	407.9 ^a^	213.6 ^a^	1329 ^a^
	Sum-Aut19	149.8 ^e^	79.4 ^e^	976 ^de^	152.7 ^e^	17.62 ^c^	0.55 ^d^	7.65 ^a^	5.49 ^a^	1.99 ^c^	0.63 ^b^	332.4 ^c^	153.5 ^c^	1057 ^b^
	Win-Spr19	193.3 ^d^	94.9 ^d^	1007 ^cd^	195.1 ^d^	27.28 ^b^	0.63 ^ab^							
Average 0		215.0 ^B^	106.8 ^B^	972 ^B^	221.3 ^B^	20.95 ^B^	0.56 ^B^	7.22 ^A^	4.91 ^A^	2.31 ^A^	0.6 ^B^	370.1 ^A^	182.0 ^A^	1144 ^A^
Difference 0 vs. 367 (%)		78.6	78.9	41.8	22.1	22.2	7.1	-	-	-	36.7	−10.1	-	-

† Means followed by different lowercase letters are different by Fischer’s protected LSD test (*p* < 0.05) within a given column. ‡ Means followed by different capital letters are different by Fischer’s protected LSD test (*p* < 0.05) within a given column.

**Table 3 plants-12-02633-t003:** Means of herbage mass (HM), herbage accumulation rate (HAR), tiller density (TD), tiller weight (TW), leaf blade length (LBL), leaf blade width (LBW), number of total leaves (NTL), number of elongating leaves (NEL), number of unfolded leaves (NUL), leaf elongation rate (LER), phyllochron (Phyll.), leaf elongation time (LET), and leaf half-life (LHL), for a group of 10 *Paspalum notatum* genotypes, and this genotypes grouped according to its growth habits (upright, intermediate and prostrated).

GENOTYPE	HM	HAR	TD	TW	LBL	LBW	NTL	NEL	NUL	LER	LET	PHYLL.	LHL
	g m^−2^	mg m^−2^ °Cd^−1^	Tiller m^−2^	mg tiller^−1^	cm		Leaves tiller^−1^			mm °Cd^−1^	°Cd^−1^		
ARG	243.6 ^a†^	121.2 ^a^	1230 ^ab^	190.1 ^bc^	19.02 ^d^	0.58 ^bc^	7.52 ^bc^	5.11 ^bc^	2.41 ^a^	0.59 ^d^	323.9 ^de^	154.9 ^c^	1052 ^d^
B14	316.7 ^a^	157.1 ^a^	1180 ^ab^	261.3 ^abc^	23.29 ^abc^	0.59 ^b^	7.26 ^bcd^	4.84 ^bc^	2.42 ^a^	0.68 ^bcd^	388.6 ^ab^	176.8 ^b^	1108 ^bcd^
B7	312.5 ^a^	155.7 ^a^	1198 ^ab^	257.3 ^abc^	24.31 ^abc^	0.59 ^b^	7.04 ^cd^	4.99 ^bc^	2.08 ^c^	0.74 ^ab^	366.2 ^bc^	173.8 ^bc^	1155 ^ab^
BOY	332.9 ^a^	165.1 ^a^	1201 ^ab^	259.2 ^abc^	26.29 ^a^	0.54 ^cd^	7.3 ^bcd^	4.97 ^bc^	2.33 ^ab^	0.77 ^ab^	360.5 ^bc^	185.3 ^b^	1188 ^a^
C11	267.2 ^a^	132.8 ^a^	1009 ^c^	241.4 ^abc^	21.71 ^bc^	0.66 ^a^	7.07 ^cd^	4.75 ^c^	2.32 ^ab^	0.69 ^bc^	346.5 ^cd^	178.1 ^b^	1117 ^bc^
C14	390.9 ^a^	194.7 ^a^	1259 ^bc^	305.3 ^a^	26.18 ^a^	0.59 ^b^	6.93 ^d^	4.70 ^c^	2.22 ^bc^	0.82 ^a^	361.3 ^bc^	185.3 ^ab^	1176 ^a^
F44	252.2 ^a^	125.8 ^a^	1231 ^ab^	186.2 ^c^	19.01 ^d^	0.58 ^b^	8.22 ^a^	5.79 ^a^	2.43 ^a^	0.68 ^bcd^	293.6 ^e^	153.9 ^c^	1077 ^cd^
G37	292.0 ^a^	145.4 ^a^	1101 ^bc^	266.3 ^abc^	23.44 ^bc^	0.58 ^b^	7.12 ^cd^	4.67 ^c^	2.46 ^a^	0.61 ^cd^	412.0 ^a^	187.7 ^ab^	1145 ^ab^
I21	280.3 ^a^	140.0 ^a^	1301 ^a^	192.5 ^bc^	19.41 ^cd^	0.49 ^d^	7.68 ^b^	5.24 ^b^	2.44 ^a^	0.68 ^bcd^	301.6 ^e^	134.0 ^d^	1101 ^bcd^
I7	304.7 ^a^	151.8 ^a^	1028 ^c^	277.1 ^ab^	25.52 ^a^	0.59 ^bc^	6.13 ^e^	3.93 ^d^	2.20 ^bc^	0.80 ^a^	359.3 ^bc^	204.5 ^a^	1144 ^ab^
UPRIGHT	316.7 ^A‡^	157.6 ^A^	1142 ^B^	264.5 ^A^	26.0 ^A^	0.57 ^B^	6.83 ^B^	4.63 ^B^	2.20 ^B^	0.77 ^A^	362.0 ^A^	187.8 ^A^	1162 ^A^
INTERMEDIATE	316.7 ^A^	157.5 ^A^	1139 ^B^	268.6 ^A^	23.6 ^B^	0.60 ^A^	7.09 ^B^	4.74 ^B^	2.36 ^A^	0.70 ^B^	377.1 ^A^	181.8 ^A^	1136 ^A^
PROSTRATE	258.7 ^B^	129.0 ^B^	1253 ^A^	189.6 ^B^	19.1 ^C^	0.55 ^C^	7.81 ^A^	5.38 ^A^	2.43 ^A^	0.65 ^B^	306.4 ^B^	147.8 ^B^	1077 ^B^

† Means followed by different lowercase letters are different by Fischer’s protected LSD test (*p* < 0.05) within a given column. ‡ Means followed by different capital letters are different by Fischer’s protected LSD test (*p* < 0.05) within a given column.

## Data Availability

Not applicable.

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
