# Peer review of "Structural and Morphogenetic Characteristics in Paspalum notatum: Responses to Nitrogen Fertilization, Season, and Genotype"

_plants, 2023, doi:10.3390/plants12142633_

Round 1

Reviewer 1 Report

The authors propose a manuscript titled “Structural and morphogenetic characteristics and the relationship with herbage mass changes in Paspalum notatum”.

I suggest the following changes:

Materials and Methods: The details of physical and chemical properties of soil should be added.

Results: Please provide ANOVA tables and combined ANOVA tables over years with the mean squares and the significance with ** or * for 99% and 95% significance, respectively

References should be presented according the journal's instructions

Minor editing of English language required

Author Response

Reviewer 1:

The authors propose a manuscript titled “Structural and morphogenetic characteristics and the relationship with herbage mass changes in Paspalum notatum”.

I suggest the following changes:

Materials and Methods: The details of physical and chemical properties of soil should be added.

Physical and chemical properties were added to the M&M section. Please see L 568-572.

Results: Please provide ANOVA tables and combined ANOVA tables over years with the mean

squares and the significance with ** or * for 99% and 95% significance, respectively

A table containing sources of variation and significances was included. Please see Table 2. The statistical software used did not provide mean squares values for linear mixed models; however, we think that Table 2 will be helpful for easy comprehension of the results.

References should be presented according the journal's instructions

Modifications were done to fit the journal`s instructions.

Reviewer 2 Report

This manuscript reports on an elegant field study that was conducted to evaluate N rate x seasonal x growth habit/genotype interactions in Paspalum. The study followed a well designed and straightforward methodology that included intensive data collection on several growth parameters of ten distinct Paspalum genotypes during different parts of the year. The manuscript is overall well written and the results are well presented.

Comments on the manuscript include,

* Perhaps the manuscript should include a Table, with a list of acronyms. This may help the reader to follow the terminology used throughout the text.

* Should the text use the word biomass, instead of mass? The text should clarify from the onset that plant mass refers to dry weight biomass (perhaps include the definition of this term, mass or biomass, in the suggested table of acronyms)

* Perhaps highlight the key or most significant correlations with total herbage of biomass yields. Which growth analysis parameters had the best correlations with total biomass accumulation?

Some repetition is observed throughout the text with respect to listing both the name of the authors and the number of the citations, e.g. “Pakiding and Hirata [52]”. This is OK in some instances, but in many cases the citation number alone (e.g. [54]), may suffice. Suggestions on providing only the citation numbers without including the name of the authors are provided in the attached copy of the manuscript on Lines L 433-441. This would improve the flow and readability of the text.

* Was the Component Analysis included in the Discussion section? If not, is it necessary in the manuscript?

What was learned from this study that was not previously known, or did the research simply confirm what previous research had found? In the Discussion it would be valuable to highlight new understandings revealed by this study in terms of genotype/growth habit x N x seasonal interactions, in terms of future breeding goals, and in terms of possible management strategies (as already properly indicated in the Conclusions). Any specific recommendations in terms of the most important selection traits, in terms of future breeding objectives? Would different traits be selected for different forage management production systems, breeding goals, or different climatic conditions (e.g for areas that experience extended droughts or later rains in the season vs those that receive adequate rainfall)?

* In the Conclusions, perhaps indicate directions for future research. What about the effect of location, soil type, or response to plant stress during different stages of the growth cycle— on the growth parameters included on the study? Would similar results be observed under different locations or environmental conditions, e.g. under water deficits?

Additional comments on the text include,

L 343-346, Please rephrase, unclear.

L 512, Briefly describe the history of the field. Was it previously under pasture? Did it have a history of regular N applications?

L 523-524, What was the source of N fertilizer used?

L 537, Briefly describe the environmental and crop growing conditions during the experimental period, to supplement the information provided in Fig. 3. Were precipitation and weather conditions normal for that region (average year growing conditions)? Was plant growth adequate, as observed under normal or average-year growing conditions?

L 578, Is the lsd test used in the analysis appropriate as a multiple comparisons means separation test (such as the Fisher’s multiple comparison lsd test)? Please clarify.

Minor edit suggestions are included in the attached copy of the manuscript.

////

The manuscript is overall well organized and written. Minor edit suggestions are included in the attached copy of the manuscript.

Author Response

This manuscript reports on an elegant field study that was conducted to evaluate N rate x seasonal x growth habit/genotype interactions in Paspalum. The study followed a well-designed and straightforward methodology that included intensive data collection on several growth parameters of ten distinct Paspalum genotypes during different parts of the year. The manuscript is overall well written and the results are well presented.

Comments on the manuscript include,

* Perhaps the manuscript should include a Table, with a list of acronyms. This may help the reader to follow the terminology used throughout the text.

A table including variables, acronyms and measure units were included. Please see Table 1.

* Should the text use the word biomass, instead of mass? The text should clarify from the onset that plant mass refers to dry weight biomass (perhaps include the definition of this term, mass or biomass, in the suggested table of acronyms)

We prefer to maintain the term herbage mass. We refer to herbage mass as “the above-ground biomass of herbaceous plants per unit area of land above a defined reference level”. Please see: Allen V.G., C. Batello, E.J. Berretta, J. Hodgson, M. Kothmann, X. Li, J. McIvor, J. Milne, C. Morris, A. Peeters and M. Sanderson (2011) An international terminology for grazing lands and grazing animals. Grass and Forage Science, 66, 2–28.

* Perhaps highlight the key or most significant correlations with total herbage of biomass yields. Which growth analysis parameters had the best correlations with total biomass accumulation?

A sentence was added in the Discussion section. Please see lines 487-495

*  Some repetition is observed throughout the text with respect to listing both the name of the authors and the number of the citations, e.g. “Pakiding and Hirata [52]”. This is OK in some instances, but in many cases the citation number alone (e.g. [54]), may suffice. Suggestions on providing only the citation numbers without including the name of the authors are provided in the attached copy of the manuscript on Lines L 433-441. This would improve the flow and readability of the text.

Modifications were done following the reviewer’s recommendation.

* Was the Component Analysis included in the Discussion section? If not, is it necessary in the manuscript?

A paragraph discussing correlations and PCA was added in the Discussion section. Please see L 487-495.

* What was learned from this study that was not previously known, or did the research simply confirm what previous research had found? In the Discussion it would be valuable to highlight new understandings revealed by this study in terms of genotype/growth habit x N x seasonal interactions, in terms of future breeding goals, and in terms of possible management strategies (as already properly indicated in the Conclusions). Any specific recommendations in terms of the most important selection traits, in terms of future breeding objectives? Would different traits be selected for different forage management production systems, breeding goals, or different climatic conditions (e.g for areas that experience extended droughts or later rains in the season vs those that receive adequate rainfall)?

A sentence and a paragraph were added in the Discussion section. Please see L 362-364, and 511-522.

* In the Conclusions, perhaps indicate directions for future research. What about the effect of location, soil type, or response to plant stress during different stages of the growth cycle— on the growth parameters included on the study? Would similar results be observed under different locations or environmental conditions, e.g. under water deficits?

A sentence was added. Please see L 560-562

Additional comments on the text include,

L 343-346, Please rephrase, unclear.

Done.

L 512, Briefly describe the history of the field. Was it previously under pasture? Did it have a history of regular N applications?

A sentence was included in the M&M section. Please see lines 568-572

L 523-524, What was the source of N fertilizer used?

The N-source was urea. A sentence was included in the M&M section. Please see lines 585-588.

L 537, Briefly describe the environmental and crop growing conditions during the experimental period, to supplement the information provided in Fig. 3. Were precipitation and weather conditions normal for that region (average year growing conditions)? Was plant growth adequate, as observed under normal or average-year growing conditions?

A sentence was included in the Results section. Please see lines 286-290.

L 578, Is the lsd test used in the analysis appropriate as a multiple comparisons means separation test (such as the Fisher’s multiple comparison lsd test)? Please clarify.

We used a Fisher`s protected LSD test (Ott and Longneker 2001) performing a correction of error type I (Benjamini and Hochberg 1995). The statistical software provides this correction alternative. A sentence containing this explanation was included in M&M. Please see lines 644-645.

Minor edit suggestions are included in the attached copy of the manuscript.

All the reviewer`s edit suggestions were done.

Reviewer 3 Report

The manuscript is well written.A lot work has been done by authors with interesting conclusions about the management of Paspalum notatum pastures.

I have two main comments.

The first is about the title. It is not very informative. It does not reflect the work of the manuscript. Fertilization is an important part of the research and has to be refer in the title.

The second is about the material and methods. The authors have to clarify few points that are not clear.

1.       My suggestions are indicated in the accompanying document

Author Response

Reviewer 3:

The manuscript is well written.A lot work has been done by authors with interesting conclusions about the management of Paspalum notatum pastures.

I have two main comments.

The first is about the title. It is not very informative. It does not reflect the work of the manuscript. Fertilization is an important part of the research and has to be refer in the title.

The title was changed following the reviewer suggestion. The propose title is “Structural and morphogenetic characteristics in Paspalum notatum: responses to nitrogen fertilization, season, and genotype”

The second is about the material and methods. The authors have to clarify few points that are not clear.

Sentences were added to the manuscript in order to clarify the methodology in the section M&M.

My suggestions are indicated in the accompanying document

All suggested editions were added.

Round 2

Reviewer 1 Report

Table 2:  source of variation, combined significance is crucial for the statistic analysis. Please add the degrees of freedom, (df) and the Mean Squares (MS) regarding the model of analysis used.  

 Minor editing of English language required

Author Response

Dear Reviewer, 

Table 2 was modified as requested. Mean squares and degrees of freedom were added. Since the table was too large, it was included as a supplementary material (Table S1).

Thank you,

Carlos Acuña

Round 3

Reviewer 1 Report

The authors have provided a revised version addressing my comments, therefore the manuscript has been sufficiently improved to warrant publication in plants. 

Minor editing of English language required